# Nematic-to-Isotropic Phase Transition in Poly(*L*-Lactide) with Addition of Cyclodextrin during Abiotic Degradation Study

**DOI:** 10.3390/ijms23147693

**Published:** 2022-07-12

**Authors:** Joanna Rydz, Khadar Duale, Henryk Janeczek, Wanda Sikorska, Andrzej Marcinkowski, Marta Musioł, Marcin Godzierz, Aleksandra Kordyka, Michał Sobota, Cristian Peptu, Neli Koseva, Marek Kowalczuk

**Affiliations:** 1Centre of Polymer and Carbon Materials, Polish Academy of Sciences, M. Curie-Skłodowska 34, 41-819 Zabrze, Poland; kduale@cmpw-pan.edu.pl (K.D.); hjaneczek@cmpw-pan.edu.pl (H.J.); wsikorska@cmpw-pan.edu.pl (W.S.); amarcinkowski@cmpw-pan.edu.pl (A.M.); mmusiol@cmpw-pan.edu.pl (M.M.); mgodzierz@cmpw-pan.edu.pl (M.G.); akordyka@cmpw-pan.edu.pl (A.K.); msobota@cmpw-pan.edu.pl (M.S.); cchpmk@poczta.ck.gliwice.pl (M.K.); 2Polish-Romanian Laboratory ADVAPOL, M. Curie-Skłodowska 34, 41-819 Zabrze, Poland and Aleea Grigore Ghica Voda, 41A, 700487 Iasi, Romania; cristian.peptu@icmpp.ro; 3Bulgarian-Polish Laboratory COPOLYMAT, Akad. Georgi Bonchev Str., Bl. 103A, 1113 Sofia, Bulgaria and M. Curie-Skłodowska 34, 41-819 Zabrze, Poland; koseva@polymer.bas.bg; 4“Petru Poni” Institute of Macromolecular Chemistry, Aleea Grigore Ghica Voda, 41A, 700487 Iasi, Romania; 5Institute of Polymers, Bulgarian Academy of Sciences, Akad. Georgi Bonchev Str., Bl. 103A, 1113 Sofia, Bulgaria; 6Faculty of Science and Engineering, School of Science, University of Wolverhampton, Wulfruna Street, Wolverhampton WV1 1LY, UK

**Keywords:** poly(*L*-lactide), cyclodextrin, liquid crystal, chiral nematic, tailor-made properties

## Abstract

Poly(*L*-lactide) is capable of self-assembly into a nematic mesophase under the influence of temperature and mechanical stresses. Therefore, subsequent poly(*L*-lactide) films were obtained and characterized, showing nematic liquid crystal properties both before and after degradation. Herein, we present that, by introducing β-cyclodextrin into the polymer matrix, it is possible to obtain a chiral nematic mesophase during pressing, regardless of temperature and time. The obtained poly(*L*-lactide) films exhibiting liquid crystal properties were subjected to degradation tests and the influence of degradation on these properties was determined. Thermotropic phase behavior was investigated using polarized optical microscopy, X-ray diffraction, and differential scanning calorimetry. The degradation process demonstrated an influence on the liquid crystal properties of pressed polymer films. The colored planar texture of the chiral nematic mesophase, which was not observed prior to degradation in films without the addition of β-cyclodextrin, appeared after incubation in water as a result of the entrapment of degradation products in the polymer matrix. These unusual tailor-made properties, obtained in liquid crystals in (bio)degradable polymers using a simple method, demonstrate the potential for advanced photonic applications.

## 1. Introduction

The inspiration to create new functional materials is not only to increase their efficiency, but also their environmental friendliness. (Bio)degradable polymers, as alternatives to non-degradable conventional plastics, are of great importance for emerging applications, environmental protection, and sustainable development.

Cyclodextrins (CDs) are cyclic saccharides that can consist of six (*α*-CD), seven (β-CD), eight (*γ*-CD), or more glucose units linked by *α*-1,4 bonds. CDs are compounds that occur naturally; *γ*-CD is the most flexible, whereas *α*-CD is the most rigid. They can be obtained both at neutral pH and moderate temperatures from hydrolyzed starch or similar substrates by the action of glucosyl transferase enzymes, and also synthetically. CDs are shaped like a hollow truncated cone. Since the hydrogen and oxygen atoms of the glycosidic linkages face the inner side and the hydroxy groups are located on the outer side, CDs have both a hydrophobic cavity and a hydrophilic outer surface, making them water-soluble and allowing the formation of inclusion complexes with various appropriately sized non-polar organic molecules. This provides a wide range of applications as molecular cages in the pharmaceutical, agrochemical, food, and cosmetic industries [1,2]. For the environment, CDs play a very important role in the solubilizing of oligomers, enriching and removing organic contaminants, and heavy metals from the soil, water, and atmosphere. In addition, CDs have the ability to increase the solubility of chemical compounds containing carbon and hydrogen in their structure, which can improve biodegradability and bioremediation, and reduce the toxicity of plastics. This leads to an increase in plant growth and the number of microorganisms, thus minimizing the share of toxic material in the environment [3].

The introduction of water-soluble CDs into the polymers not only improves their hydrophilicity, but also affects their crystallization, degradation, and thermostability, due to the fact that hydrophobic cavities limit the motion of macromolecular chains and increase the degree of cross-linking [4]. CDs form water-insoluble crystalline host-guest inclusion complexes with many (bio)degradable polyesters, such as polylactide (PLA), poly(*ε*-caprolactone) (PCL), and polyhydroxyalkanoate (PHA). Moreover, direct addition of CDs (not as an inclusion complex) to poly(3-hydroxybutyrate) (PHB) or PLA can increase nucleation and promote PHB and PLA crystallization, in addition to increasing the processability and heat resistance of these polyesters, depending on the amount of CD used [5,6,7,8]. The properties of the PLA composite film with the PLA/β-CD inclusion complex have been compared with the PLA composite film prepared by the simple addition of β-CD. The addition of the PLA/β-CD inclusion complex to the PLA composite film resulted in significantly increased thermal expansion stability of the film. However, the mechanical and barrier properties deteriorated; this also occurred for the film to which β-CD was added due to the behavior of the complex as filler in the matrix of PLA composites [9].

The solubility of β-CD in water is low (18.4 g/L) compared to that of *α*-CD and *γ*-CD (129.5 and 249.2 g/L, respectively) [10]. Methylation of β-CD, regardless of the degree of methylation, improves its water solubility by up to 50 times compared to unsubstituted β-CD. Randomly methylated β-CD (RM-β-CD) is a lipophilic derivative, although it is soluble in water and has good binding capacity. In addition to its use in various drug formulations, RM-β-CD also has non-pharmaceutical applications, for example, reducing the viscosity of emulsion-type wall paints [11,12]. It was found that the addition of methyl-β-CD (50–83 wt%) to PLA films cast from chloroform lowers the crystallinity and enhances the mobility of the polymer. The conclusion was that methyl-β-CD had a high affinity for PLA and the addition of methyl-β-CD increased the amorphous regions of PLA enhancing the drawability [13]. In contrast, the addition of smaller amounts of CD to the PLA (up to 30 wt% for β-CD) caused a nucleation effect [9].

Native CD molecules are not capable of forming thermotropic liquid crystals (LCs) [14]. As LCs have some of the typical properties of liquids (regarding their mobility at room temperature, RT) and some properties of solid crystals, they are an intermediate state between crystal and liquid, and therefore said to be mesophases. In a liquid, the molecules are disordered, which means that they have neither a specific position nor a specific orientation or direction (isotropy). On the other hand, in LCs, although the molecules still do not have fixed positions and remain disordered, as in a liquid in a horizontal position, the molecules can also be orientated, as in a crystal, along a vertical direction (anisotropy) [15,16]. Consequently, the molecules that make up the LCs are mesogens that induce structural order: rod-like (calamitic), disk-like (discotic), or bent-core (banana) individuals that exhibit LC properties [17].

Some of the CD derivatives obtained are thermotropic LCs within certain temperature ranges. CD inclusion complexes may also exhibit LC properties in an aqueous medium [18]. CD-based thermotropic LC systems can be divided into LCs based on CD derivatives having aliphatic chains and polar groups or containing mesogenic groups, in addition to CD-based LC polymers [14]. Many surfactants can form LC structures with CDs. An ordered LC structure with layers in which the molecules are positionally ordered in one direction was obtained from polyanionic hyaluronan with entrapped cationic surfactant. The surfactant/*γ*-CD inclusion complex acted as a bifunctional electrostatic cross-linker. This highly ordered complex lamellar structure is typical of long-chain guest inclusion complexes [19]. However, most of the CD complexes exhibiting LC properties are lyotropic, with hydrophilic and hydrophobic domains that can aggregate adjacent to each other to form an anisotropic system have a long-range order [18].

Polymers can be converted into mesophases under elevated pressures [20]. Pressure-induced liquid crystallinity is simply a consequence of the reduced molar volume (relative to the isotropic state) that characterizes fluids having a long-range orientational order. The same phenomenon can be used to induce LC in thermotropic polyesters such as PLLA or polycaprolactone [21,22,23]. Mesophases have been induced in non-mesogenic PLLA by pressing at a controlled temperature. This process was also accompanied by the formation of multiple morphologies [23,24,25]. The formation mechanism of the mesophase obtained by pressing is likely related to the limited degrees of freedom of polymer chains at temperatures close to the glass transition temperature (*T_g_*). The frozen glassy structure of the polymer below *T_g_* is limited to vibrational motion only, whereas heating to *T_g_* allows limited rotational and translational motion as the polymer approaches a rubbery state. Beyond the *T_g_* region, full rotational and translational motion is possible. Thus, under external stress and temperature, this limited rotational and translational motion near *T_g_* results in a nematic-like orientation as the polymer plastically deforms. Reversible and irreversible nematic-to-isotropic phase transition is connected to the motion of the polymer chains. The irreversible transitions are associated with slower kinetic processes related to the rotational and translational motion of polymer chains, whereas the reversible transitions are associated with faster kinetic processes, which include vibrational motion, and are directly related to the heat capacity of the substance. Pressurizing the polymer can increase the free volume of the material as the result of plastic defects. The increased free volume from these defects influences the activation of molecular mobility [22,26,27,28,29,30].

This study examined the hydrolytic degradation of PLLA films before and after pressing, and pressed PLLA films with the addition of RM-β-CD at 50 and 70 °C. The effect of pressing and the presence of CD on the degree of degradation of the obtained films were investigated. The influence of the films’ degradation on the LC properties was also studied. The progress of the material hydrolytic degradation was assessed by failure analysis of the examined materials (macro- and microscopic observations of the films), thermal properties, molar mass, and molar-mass dispersity changes during the conducted experiments. The main novelty of the presented research is the combination of unique LC properties with the (bio)degradability of PLLA, and the demonstration that the degradation products act as a nucleation agent that triggers the colored planar texture of the chiral nematic (N*) mesophase.

## 2. Results and Discussion

### 2.1. Degradation Study

The hydrolytic degradation of initial PLLA rigid film and pressed PLLA and PLLA/RM-β-CD films, at the temperatures of 50 and 70 °C was carried out in laboratory conditions for a period of 70 days. A decrease in the transparency of the tested PLLA films was observed from the beginning of degradation at both temperatures, as a result of molecular reorganization or an increase in irregularities, which may result from the accelerated formation of new spherulites [31,32]. The effect was the most pronounced for the PLLA film after 21 days of degradation at 50 °C (Figure 1).

The degradation process in water caused a continuous decrease in the molar mass of all samples from the beginning of the experiment at 70 °C and after the 3rd day of degradation at 50 °C. By day 3, if a molar-mass change occurred, it was within the experimental error of the measurements (approx. 5–10%). After 21 days of degradation at 70 °C, all samples disintegrated. The amorphous initial PLLA degraded slightly faster in comparison to both pressed films: 16 ± 0.7% after 7 days at 50 °C and 10 ± 1.4% after 3 days at 70 °C (at the most significant change) (Figure 2 and Table 1). This trend was confirmed by the analysis of the pH of the solution (data not shown).

A slight effect of RM-β-CD (5% acceleration) on degradation was visible at 50 °C, where slight changes were easier to observe than at 70 °C due to the slower process [32]. This acceleration may be due to the presence of a fine powder such as RM-β-CD, which facilitates the access of water, and a more hydrophilic outer surface of CDs than PLA. Comparison of average contact angle values for pressed PLLA and PLLA/RM-β-CD films (82.64 and 72.56, respectively) confirmed that the pressed PLLA film surface was more hydrophobic.

The thermal properties of the initial PLLA rigid film and pressed PLLA and PLLA/RM-β-CD films, during degradation were evaluated by the DSC method. The results of the thermal analysis supplemented with changes in the molar mass are presented in Table 2. 

At the first heating run with a rate of 20 °C·min^−1^ in which the thermal history was suppressed, the glass transition overlapped with the structural relaxation was observed for the initial amorphous PLLA rigid film [33]. RM-β-CD contains crystallization water, which is lost at a temperature of 20–120 °C [34]. As the water content (acting as a plasticizer) in amorphous RM-β-CD increases, *T_g_* decreases [35]; therefore, the analyses of RM-β-CD were carried out in hermetically sealed pans. DSC analysis shows that the RM-β-CD powder was amorphous, with only *T_g_* = 72.7 °C and Δ*cp* = 0.39 J·g^−1^·°C in the II-heating run after rapid cooling (data not shown), whereas the pressed PLLA and PLLA/RM-β-CD films were partially ordered (see Δ*H_m_* and Δ*H_cc_* in Table 2). However, not only the addition of RM-β-CD to the PLLA film, but also the pressing process [25], resulted in an increase in the order of the polymer chains (the absolute value of the difference between Δ*H_m_* and Δ*H_cc_* for individual polymers: initial PLLA < pressed PLLA < pressed PLLA/RM-β-CD). The addition of RM-β-CD led to a higher melting temperature (*T_m_*) and Δ*H_m_* of pressed PLLA/RM-β-CD compared to pressed PLLA film, in addition to a slightly higher *T_g_*, as CDs are considered to be a nucleation agent that increases the order of the film [9]. It is well documented that the mechanical properties of PLA are strongly dependent on the structures and crystal morphology, and that the addition of a nucleating agent increases the crystallization because this lowers the free surface energy barrier to nucleation [36]. During the first heating run, a broad exothermic peak at 122.0 °C with Δ*H_cc_* = 9.17 J·g^−1^ and a melting endotherm at 150.3 °C with enthalpy Δ*H_m_ =* 13.77 J·g^−1^ due to ordering were detected. The glass transition temperature at 62.1 °C was taken from the second heating run at a rate of 20 °C·min^−1^ (Table 2).

It is well known that the *T_g_* of a polymer depends on the length of the chains [37]. During the degradation process, after predetermined incubation times, the *T_g_* decreased in the following order: initial PLLA > pressed PLLA > PLLA/RM-β-CD. Degradation causes changes in crystallization potential and, consequently, this leads to changes in melting. Strong dependence of Δ*H_m_* on degradation time was observed. As the orderliness of the polymer increases as a result of degradation, Δ*H_m_* increases, and *T_m_* decreases. However, as the order of the polymer chains increased prior to degradation in the order initial PLLA > pressed PLLA > PLLA/RM-β-CD, the increase in orderliness during slower degradation at 50 °C for pressed PLLA/RM-β-CD films was not that significant. This is probably due to the nature of the degradation process, which occurs in a privileged manner in the amorphous regions of the sample, causing faster degradation of this phase [38]. This is because an increase in orderliness results in a higher diffusion resistance to transport and the ratio of the effective diffusion coefficient (depending on the concentration or upstream pressure) to the diffusion coefficient (expressed as length^2^·time^−1^) decreases [39]. In the case of faster degradation at 70 °C, an increase in orderliness is observed first until the sample disintegrates, i.e., until the 21st day of incubation (the amorphous phase is degraded, and the degradation products diffuse more slowly into the environment), and followed by a decrease in Δ*H_m_*, caused by both greater washing out of degradation products to the environment and the beginning of the degradation of the crystalline phase.

### 2.2. Liquid Crystal Properties

The N* mesophase was obtained under the influence of temperature and mechanical stresses caused by pressing. First, the initial PLLA rigid film was heated to 110 °C (from RT to a temperature above its *T_g_* = 60 °C and below its *T_m_* = 152 °C; see Table 2); the polymer was then subjected to a pressure of 5 tons for about 1 min while the temperature was maintained at 110 °C; and finally, the PLLA film was cooled to RT without maintaining the elevated pressure. When the film sample was cooled to below *T_g_*, the polymer retained the structure of the nematic mesophase [24].

The phase behavior of samples was examined by observation of optical textures using POM (Figure 3). The characteristics of individual textures are detailed in [25].

It was previously found [25] that non-mesogenic thermoplastic PLLA rigid film is capable of self-assembly into a nematic mesophase under the influence of temperature, pressure, and time [20,21,40]. The POM observation of pressed PLLA/RM-β-CD film revealed that, by incorporating a small amount (0.5 wt%) of RM-β-CD into the polymer matrix, a colored planar texture of the N* mesophase can be obtained (Figure 3B, compare with further Figure 7_0) [41,42,43]. Additionally, this effect was observed after degradation of the pressed PLLA, especially after 1 day of degradation (compare with further Figure 7 and Figure 9), suggesting that the degradation products and CDs in the form of fine powder acted as a nucleation agent and triggered the colored planar texture of the N* mesophase. RM-β-CD acted as a nucleation agent (increasing nucleation density) that accelerated ordering and facilitated the twisting of the molecules perpendicular to the axis of rotation. The coexistence of domains having a colored planar texture (Figure 3B) and a schlieren texture of the N* mesophase (Figure 3C–F) was observed, especially at the edges of the films, where the RM-β-CD distribution can be more uneven due to more difficult distribution; hence, the heterogeneity resulting from more disperse nucleation. The bulk heterogeneity of the material is typical for polymers having LC properties [44].

The thermal behavior of the obtained materials was investigated in order to confirm the nematic mesophase. Representative POM photomicrographs and DSC traces of the pressed PLLA/RM-β-CD film during the first cooling run and second heating run at a rate of 10 °C·min^−1^ are presented in Figure 4 and Figure 5. The film sample was first heated from 10 to 200 °C at a rate of 20 °C·min^−1^ to eliminate the effect of thermal history.

The nematic phase has been confirmed by the POM observation of optical textures and DSC studies together, and the most important evidence of mesogenic behavior of the pressed PLLA/RM-β-CD is the observation of phase transitions as a function of temperature under POM equipped with a hot-stage. A material in which phase transition was observed as a function of temperature formed the mesophase exhibiting LC properties. A reversible nematic to isotropic transition is not observed in the case of a stressed polymer films. A detailed discussion of all aspects of the LC phenomenon in PLLA pressed films is presented in an Open Access publication [25].

In the first cooling run and second heating run at 10 °C·min^−1^, the pressed PLLA/RM-β-CD film exhibited only *T_g_* and mesophase-to-isotropic state transitions (Figure 4). Pressed PLLA/RM-β-CD exhibited a clearing point at 154.0 °C (Δ*H_NI_* = 0.22 J·g^−1^) and isotropic-to-mesophase state transitions at 148.0 °C (Δ*H_NI_* = 0.2 J·g^−1^) during heating and cooling runs, respectively, which were also confirmed under POM with a hot-stage (see Figure 5). The enthalpy of phase transition during the cooling and heating process was similar.

Figure 5A shows the nematic mesophase of pressed PLLA/RM-β-CD under crossed polarizers at 146 °C. By raising the temperature to 153 °C, the nematic mesophase lost birefringence and was transformed into an isotropic phase and the texture turned dark (Figure 5B). After cooling the isotopic phase to 148 °C, the nematic texture formed again (Figure 5C). After further heating and cooling, this effect was also observed again. The POM observation of the optical texture of the mesophase (the colored planar Grandjean texture), which was first heated and then cooled, in addition to the reversible nematic-to-isotropic phase transition and low enthalpy of this transition, (Δ*H_NI_* = 0.2 ± 0.01 J·g^−1^), found in DSC experiments, suggests that pressed PLLA/RM-β-CD forms a stable N* mesophase (Figure 4 and Figure 5).

In addition, an X-ray diffraction analysis of the initial PLLA rigid film, RM-β-CD powder, and pressed PLLA and PLLA/RM-β-CD films was performed to confirm the presence of the nematic mesophase and to investigate the effect of RM-β-CD on the PLLA films’ crystalline phase (Figure 6).

As previously described, a broad scattering pattern was observed for the initial PLLA rigid film attributed to the reflection of amorphous phase in the 2θ region of 8–26°, and, on the contrary for the pressed PLLA film, sharp diffraction peaks typical of the PLLA crystalline profile, indicating a reduced amorphous fraction [25]. X-ray diffractograms of RM-β-CD showed two wide peaks in the 2θ region of 5–15° and 15–30°, which also confirmed its amorphous nature [45].

The diffractogram of the pressed PLLA/RM-β-CD was found to differ significantly from that of the initial PLLA rigid film and RM-β-CD powder, and only slightly from pressed PLLA. A slight peak shift was detected, from 16.78° 2θ for pressed PLLA film to 16.84° 2θ for PLLA/RM-β-CD, which suggests the presence of residual stress.

The analysis of the X-ray pattern of pressed PLLA/RM-β-CD at 25 °C (Figure 6) shows the presence of the reflections at wide angles, confirming a crystalline fraction and an amorphous halo in the 2θ region of 5–27° on the flat film diffractogram, which arises from the interchain distances, and is probably responsible for the existence of the N* mesophase [46]. It is known that a nematic mesophase only shows a diffuse halo in a high-angle region [47,48]. The point here is that the LC regions can cover all or only a part of the material. Earlier investigations [25] showed that the material was amorphous when processed up to 100 °C. The material preparation temperature between 110 to 130 °C (as for pressed PLLA and PLLA/RM-β-CD films) accelerated order, making the material heterogeneous. At the processing temperature of 140 °C, the pressed PLLA films no longer showed LC properties as they became crystalline.

Table 3 shows the crystallographic data of the pressed PLLA and PLLA/RM-β-CD films, and the pressed PLLA films, after 1 day of degradation at 50 and 70 °C. 

Compared with the model data for the reference polylactide, and with the values of pressed PLLA, the PLLA/RM-β-CD films showed a reduction in the parameter a (1.6% and 0.5 %, respectively) and the parameter c (1.9% and 2.6%, respectively) of the orthorhombic P2_1_2_1_2_1_ unit cell. It should be noted that parameter b did not change compared to reference polylactide, whereas the lattice volume changed significantly, which suggests the contribution of shear stress. However, the use of the extended Bragg’s law does not provide information about shear stress, as it calculates the mean value in the form of linear stress, which, in this case, shows a compressive nature and has a relatively high value. Another argument suggesting the contribution of shear stress is the high value of lattice strain compared to that in pressed PLLA. Moreover, after the introduction of the RM-β-CD, the residual stress changed from high tensile to average compressive.

Calculated by the peak decomposition method, the relative crystallinity of the PLLA matrix for pressed PLLA film was 1.6% [25], and that of PLLA/RM-β-CD was 6.4%. The use of β-CDs as nucleating agents for PLLA provides an increase in the orderliness of the polymer matrix. The same effect has been observed previously with talc [25]. The obtained results suggest that introduction of RM-β-CD particles into the PLLA matrix cause shear stress and induce the N* mesophase. For pressed PLLA films after degradation, the crystalline phase increased with degradation temperature, and its values were 11.5 and 16.7%, respectively, at 50 and 70 °C. For samples after degradation, the residual stress was more pronounced with the increase in the degradation temperature, but the lattice volume and parameters suggest the contribution of shear stress at lower degradation temperature (sample after 1 day of degradation at 50 °C). Most likely, at a higher temperature, a partial shear stress relaxation occurs in the form of chain straightening, resulting in higher linear stress and almost ideal lattice parameters.

### 2.3. Surface Characteristics

In the case of (bio)degradable polymers, it is important to understand the degradation and aging processes occurring in (bio)degradable polymer materials. To some extent, PLA mesophases can form naturally during the aging process [22,49]. Most polymers undergo physical aging and other structural changes over time. A process of physical aging (structural relaxation) has an additional effect on the properties of polymers. Enthalpy relaxation is the process in which a glass-like material approaches a state of thermal equilibrium through the release of free energy (enthalpy) by the realignment of structural/chain segments and, consequently, the loss of free volume, depending on the proximity of the material to its *T_g_* and the time at which it is maintained there. This process, however, can be slow, often taking months or years, depending on the proximity of the materials to the *T_g_* during storage [24,50,51]. Aging of the polymers occurs when the material is brought below the *T_g_* at a rate that prevents it from reaching thermal equilibrium before entering the slowly moving glassy state. As the material reaches the glassy state, the amorphous polymer chains are frozen, trapping excess enthalpy and free volume in their physical structure. However, over time, the material relaxes and moves towards thermal equilibrium, releasing this excess enthalpy and free volume by shifting small molecule segments and chains over time (increase in enthalpy relaxation, decrease in specific heat capacity, and increase in *T_g_* as the material relaxes or loses this free energy in its path to equilibrium through physical relaxation). During this process, a new amorphous phase can be formed, having both a *T_g_* below the original and a phase with *T_g_* higher than the original *T_g_*, which can be related to the ongoing crystallization process. Materials having enthalpy relaxation and specific heat capacity are sensitive to pressure as they decrease in response to increasing pressure [24,51,52].

During degradation of pressed PLLA film with a thread-like texture [25], especially after 1 day of degradation, a colored planar Grandjean texture of an N* mesophase was observed in POM, indicating that the degradation products trapped in the polymer matrix acted as a trigger for the N* mesophase. It is also possible that the aging phenomenon played a role.

As degradation progressed, small cracks formed as a result of water absorption and polymer erosion, which grew when contact with water led to hydrolysis [31]. The degradation products initially trapped in the polymer matrix were released into the degradation medium (water) during degradation, and the effect of the colored planar texture of the N* mesophase was diminished (Figure 7).

After 70 days of degradation at 50 °C of the initial PLLA rigid film, phase separation was observed between the polymer matrix and the degradation products of relatively low molar mass [53]. In this case, a complex melting pattern was also observed, with two main endotherms located at 132.8 and 149.8 °C (see Table 2), indicating a sample decomposition leading to different populations of PLLA crystallites with different molar-mass macromolecules [32]. It is also generally known that the overall rate of crystallization increases as the molar mass of PLLA decreases [54]. Comparison of X-ray diffraction patterns for the initial PLLA and samples after 70 days of degradation at 50 °C showed an increase in PLLA orderliness. The crystallinity of this sample was 34.5%, with a mean crystallite size of approx. 66 nm. The calculated lattice parameters for samples after 70 days of degradation at 50 °C showed a major decrease in all lattice constants (Table 3), which may increase the crystallization of lower-mass macromolecules compared to other samples. Moreover, the presence of residual stress was detected, with a mean value of 5.7 ± 0.6 MPa, visible as peak shift in the direction of lower angles. The strongest shift was detected for the (200) lattice plane, where a major contraction of the lattice constant occurred. However, peaks corresponding to the *c* axis also shifted, but with different intensities; this may suggest the presence of shear stress that can induce phase separation in the polymer matrix.

By means of the AFM and SEM techniques, the surface of the films was examined. The AFM two-dimensional (2D) and three-dimensional (3D) topography images provide information about the surface morphology, thus highlighting defects and distinguishing between phases or surface roughness parameters, such as grain size or RMS roughness (*R_q_*) and 3D roughness average (*R_a_*) [55,56].

The AFM examination of the surface of the films revealed that the non-mesogenic thermoplastic initial PLLA rigid film exhibited a molecular disorder, whereas the nematic bulk mesophase of pressed PLLA showed an arrangement having a long-range molecular order with a thread-like texture [25]. By comparison, the pressed PLLA/RM-β-CD film showed an N* mesophase having a heterogeneous surface, showing a coexistence of Grandjean and fingerprint textures of N* (see Figure 8, samples before degradation). As the degradation proceeded, the amorphous regions degraded, leaving a more ordered polymer (see Figure 8, samples after 1 and 70 days of degradation).

Surface erosion during degradation facilitated access to deeper areas of the sample, revealing areas characteristic for polymers with LC properties. Additionally, as mentioned above, the degradation products acted as a nucleation agent and resulted in the formation of the N* mesophase, which facilitated its observation (Figure 8). For example, pressed PLLA film after 1 day of degradation (Figure 8III) showed areas of clear phase separation between the rigid mesogens and the relatively soft part of the polymer; pressed PLLA film after 70 days of degradation (Figure 8II) showed a cross section through the N* structure forming an arc-like pattern; and pressed PLLA/RM-β-CD film after 70 days of degradation (Figure 8II) showed the characteristic dislocation lines [57].

Table 4 presents *R_a_*, which is the mean height calculated over the entire surface area measured, and *R_q_*, which is the square root of the surface height distribution and is considered more sensitive than *R_a_* for large deviations from the mean plane [55].

The roughness of the tested films, measured on a horizontal and vertical length scale of 0.5 µm, increased from 1.3 nm for the initial PLLA rigid film to 3.3 nm for the pressed PLLA film and 3.9 nm for pressed PLLA/RM-β-CD film, whereas the pore depth increased from 7.7 to 19.7 and 28.5 nm, respectively. This is evidence of an increase in the heterogeneity of the material surface, typical for polymers with LC properties.

Large-scale SEM images, 3 mm × 2.6 mm (I), 600 μm × 520 μm (II), and 147 μm × 127 μm (III), shown in Figure 9, respectively, show clear differences between the N and N* mesophases. In the SEM images of the N* mesophase, the agglomeration of fibrils that appear on the surface of the film can be seen much more clearly, as twisting, splitting, bending and aggregating, or intertwining [25]. Figure 9IIIB shows the characteristic dislocation lines in the planar Grandjean texture of the N* mesophase [44].

## 3. Materials and Methods

### 3.1. Materials

Random methyl-β-cyclodextrin (RM-β-CD), degree of substitution (DS) of ~12, from CycloLab Ltd. (Budapest, Hungary), was used as received. An initial PLLA rigid film with a thickness of 0.33 ± 0.02 mm, an average mass of 97 ± 2 mg, a mass-average molar mass *M_w_* = 180,000 g·mol^−1^, and molar-mass dispersity *M_w_*/*M_n_* = 2.0 (determined by gel permeation chromatography), obtained by extrusion followed by thermoforming prepared as described in reference [58], was used as the initial material for pressing.

### 3.2. PLLA Films Preparation

Neat pressed PLLA films having a thickness of 0.08 ± 0.02 mm and an average mass of 97 ± 3 mg for tests were prepared from the initial PLLA rigid films as described in reference [25] on a hydraulic press having a force of 5 tons at the temperature of the press heating plate (110 °C) for 1 min. Pressed PLLA/RM-β-CD films having 0.5 wt% of RM-β-CD, and a thickness of 0.1 ± 0.02 mm and an average mass of 99 ± 1 mg, were pressed into films as above.

### 3.3. Abiotic Degradation Study

For the hydrolytic degradation under laboratory conditions, according to the ISO norm [59], initial PLLA rigid film and pressed PLLA and PLLA/RM-β-CD films, were incubated at 50 and 70 °C (± 0.5 °C) in 30 mL screw-capped vials with an air-tight PTFE/silicone septum, containing 25 mL of demineralized water (pH = 5.8) over a period of 70 days. The degradation experiment was run in triplicate. After a predetermined degradation time, the samples were separated from the degradation medium, washed with demineralized water, and dried, first on filter paper and then under a vacuum at a temperature of 25 °C, to a constant mass. After a specified period of time (1, 3, 7, 21, 70 days) surface erosion, the samples’ molar mass, molar-mass dispersity, and thermal characteristics, in addition to the molecular structure of the samples, were determined. The molar-mass loss was determined with triplicate measurements and calculated as described elsewhere [31,60].

### 3.4. Characterization

#### 3.4.1. Polarized Optical Microscopy (POM)

The PLLA films that exhibited LC properties were observed with a Zeiss polarized optical microscope (Opton-Axioplan, Oberkochen, Germany) equipped with a Nikon Coolpix 4500 (Tokyo, Japan) color digital camera and Mettler FP82 hot plate with Mettler FP80 temperature controller. The sample was placed on a microscope slide with a cover slip, and the slide was then heated and cooled while observing the phase changes.

#### 3.4.2. Scanning Electron Microscopy (SEM)

SEM studies were performed using of a Quanta 250 FEG (FEI Company, Fremont, CA, USA) high resolution environmental scanning electron microscope operated at a 5 kV acceleration voltage. The film samples were observed without coating under low vacuum (80 Pa) using a secondary electron detector (large field detector).

#### 3.4.3. Atomic Force Microscopy (AFM)

AFM measurements were performed using a Dimension ICON AFM microscope equipped with a NanoScope V controller (BRUKER Corporation, Santa Barbara, CA, USA) operating in soft tapping mode in an air atmosphere with a standard 125 μm long and 10–15 μm high tip, and a single-crystal doped silicon cantilever having flexural stiffness of 42 N m^−1^ (Model RTESP-300, BRUKER, Camarillo, CA, USA). Images were obtained with a piezoelectric scanner having a nominal size of 85 × 85 μm. The micrographs were recorded using NanoScope Analysis 1.9 Software (BRUKER Corporation, Santa Barbara, CA, USA). The most representative images for each PLLA film were selected from three measurements taken on several different samples.

#### 3.4.4. Differential Scanning Calorimetry (DSC)

Thermal characteristics of the samples were obtained using a differential scanning calorimetry TA-DSC Q2000 apparatus (TA Instruments, Newcastle, DE, USA). The instrument was calibrated by indium having high purity. The first heating run concerned initial samples in which the thermal history was suppressed. DSC studies were carried out at a temperature from −90 to 200 °C at the rate of 20 °C·min^−1^ (I- and II-heating run) and 10 °C·min^−1^ (I-heating and cooling as well as II-heating run). All of the experiments were performed under a nitrogen atmosphere with a nitrogen flow rate of 50 mL·min^−1^, using aluminum standard sample pans. The *T_m_* was taken as the peak temperature maximum of that melting endotherm, and *T_g_* was taken as the midpoint of the heat capacity change in the sample.

#### 3.4.5. X-ray Diffraction and Residual Stress Analysis

X-ray diffraction was performed using a D8 advance diffractometer (Bruker, Karlsruhe, Germany) with a Cu-Kα cathode (*λ* = 1.54 Å). The scan rate was 1.2°·min^−1^ with a scanning step 0.02° in the range of 5° to 60° 2θ using Bragg-Brentano geometry, and residual stress analysis was performed using grazing incidence geometry with an incidence angle of 1°. All measurements were performed in triplicate. Identification of fitting phases was performed using the DIFFRAC.EVA program with the ICDD PDF#2 database, and the crystalline size, lattice strain, and lattice parameters of *P*2_1_2_1_2_1_ orthorhombic PLLA crystallites were calculated using Rietveld refinement in the TOPAS 6 program, basing on Williamson-Hall theory as described in [25].

#### 3.4.6. Gel Permeation Chromatography (GPC)

The molar mass and molar-mass dispersity of the films were determined using GPC conducted in chloroform solution at 35 °C and an eluent flow rate of 1 mL·min^−1^ with a set of two PL-gel 5 μm MIXED-C ultrahigh efficiency columns (Polymer Laboratories, Church Stretton, UK) having a mixed bed and a linear range of *M_w_* = 200–2,000,000 g·mol^−1^. A VISCOTEK VE 1122 isocratic pump (Malvern Panalytical Ltd., Malvern, UK) was used as the solvent delivery system with a Shodex SE 61 differential refractive index detector. A quantity of 10 μL of 0.3 % m·V^−1^ sample solution was injected into the system. To prepare a universal calibration curve, polystyrene standards with narrow molar-mass dispersity (Calibration Kit S-M-10, Polymer Laboratories) were used. *M_w_* of the samples was determined using OmniSEC 5.0 (Viscotek, Malvern, UK) software. The molar-mass loss was calculated using Equation (1):% molar-mass loss = [(*M_w_*_0_ − *M_wx_*)/*M_w_*_0_] × 100%(1)
where *M_w_*_0_ is the initial mass-average molar mass and *M_wx_* is the consecutive or final average molar mass.

#### 3.4.7. Contact Angle Measurements

The surface properties of the tested material were statistically measured with a CAM101 contact angle goniometer (KSV Instruments, Helsinki, Finland). The water contact angles were determined in air using the sessile-drop method. A series of images for the water drops (5 µL) were acquired for 15 s, and the average contact angle value was calculated. The contact angle measurements were made at 20 °C on dry surfaces. The average of five values of contact angles from different parts of the film samples was calculated. Two samples were used for measurements from each surface.

## 4. Conclusions

It was previously reported that, in a non-mesogenic thermoplastic polymer, LC properties similar to those of a nematic mesophase having a specific texture can be induced by exposing the polymer to pressure at a temperature close to *T_g_* and additionally introducing a fine powder (talc) [25]. The current research showed that only dark and bright domains having a thread-like texture were obtained during the preparation of films at the processing temperature of 110 °C, close to *T_m_* of PLLA (at a pressure of 5 tons for 1 min). These films were partially ordered with an irreversible monotropic nematic mesophase. In this study, it was shown that, by introducing RM-β-CD to the pressed PLLA films, a colored planar texture of an N* mesophase (with interference colors visible under POM) can also be obtained. For this texture, characteristic surface disclination lines under SEM were observed. Interestingly, this texture was also obtained during the degradation of the pressed PLLA films without the addition of CD due to the degradation products trapped in the polymer matrix. The degradation process influenced the LC properties of pressed PLLA and PLLA/RM-β-CD films. However, the progress of the degradation causes leaching of the degradation products, resulting in a reduction in this effect. The observations of mesophase textures and transitions were found to be consistent with the DSC results and the X-ray diffraction studies. The resulting enantiotropic mesophase was stable at both ambient temperature and during degradation at 50 °C, and had a low energy state. From an environmental perspective, green nucleating agents are desirable, especially for biodegradable polymers. The addition of randomly methylated β-CD acting as an environmentally friendly nucleating agent presents an effective means to increase the crystalline phase in PLLA.

## Figures and Tables

**Figure 1 ijms-23-07693-f001:**
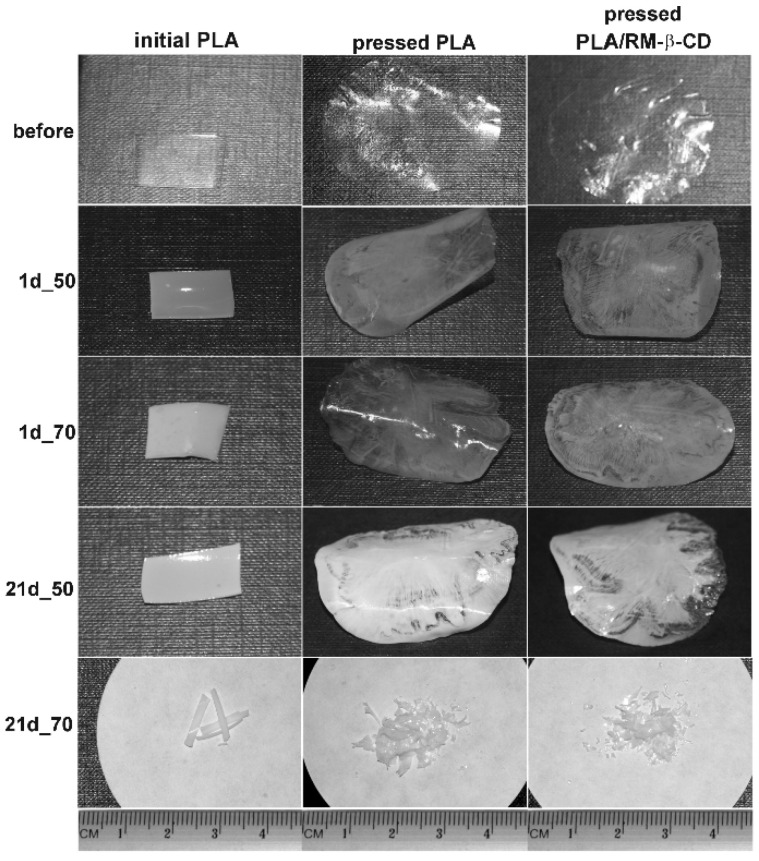
Macrographic images of the initial PLLA rigid film and pressed PLLA and PLLA/RM-β-CD films, before and after 1 (1d) and 21 (21d) days of hydrolytic degradation at 50 and 70 °C.

**Figure 2 ijms-23-07693-f002:**
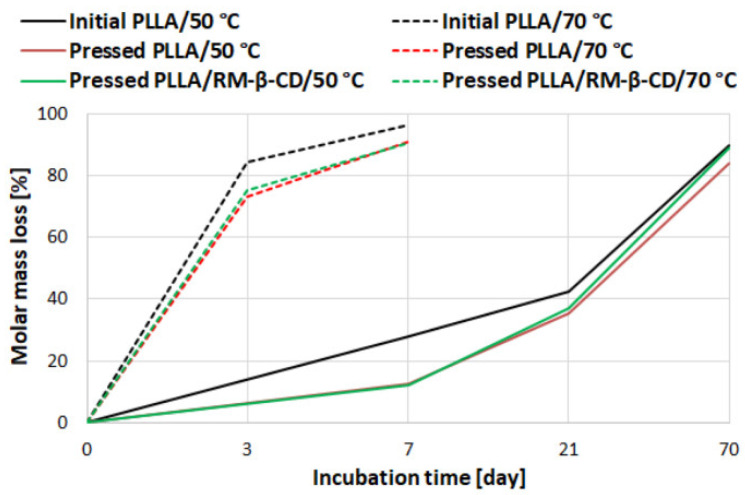
Molar-mass loss of initial PLLA rigid film and pressed PLLA and PLLA/RM-β-CD films, as a function of incubation time of the degradation test at 50 and 70 °C. The molar-mass loss is given as a percentage of the original *M_w_*.

**Figure 3 ijms-23-07693-f003:**
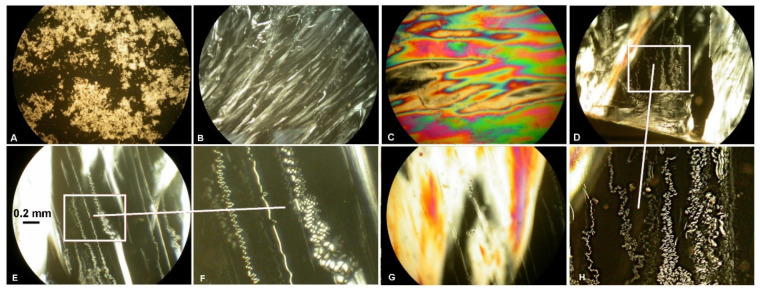
Representative photomicrographs of optical textures of the amorphous RM-β-CD (**A**), nematic mesophase of the pressed PLLA film (thread-like texture, (**B**), colored planar texture of the chiral nematic mesophase of the pressed PLLA/RM-β-CD film (**C**), nematic mesophase with a heterogeneous surface (**G**), and schlieren texture of nematic mesophase of the pressed PLLA/RM-β-CD film (**D**–**F**,**H**; (**E**,**F**)—enlarged image showing topological defects) (crossed polarizers, 25 °C, 100×).

**Figure 4 ijms-23-07693-f004:**
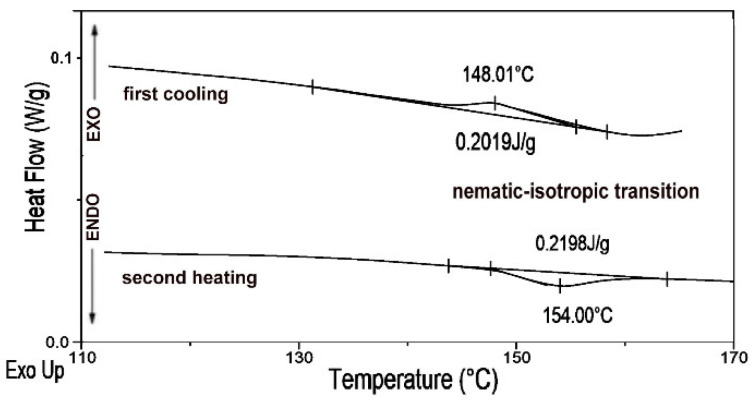
Representative DSC traces of the pressed PLLA/RM-β-CD film obtained at 10 °C·min^−1^ in the cooling run and second heating run.

**Figure 5 ijms-23-07693-f005:**
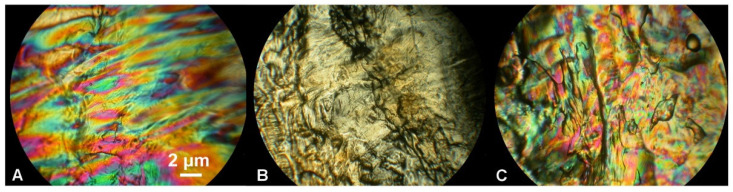
Nematic-to-isotropic phase transition of the pressed PLLA/RM-β-CD. Photomicrographs of the optical texture of the chiral nematic mesophase during heating and cooling at 146 °C (nematic mesophase (**A**)), 153 °C (isotropic phase (**B**)), and 148 °C (nematic mesophase after cooling (**C**)) (crossed polarizers, 160×).

**Figure 6 ijms-23-07693-f006:**
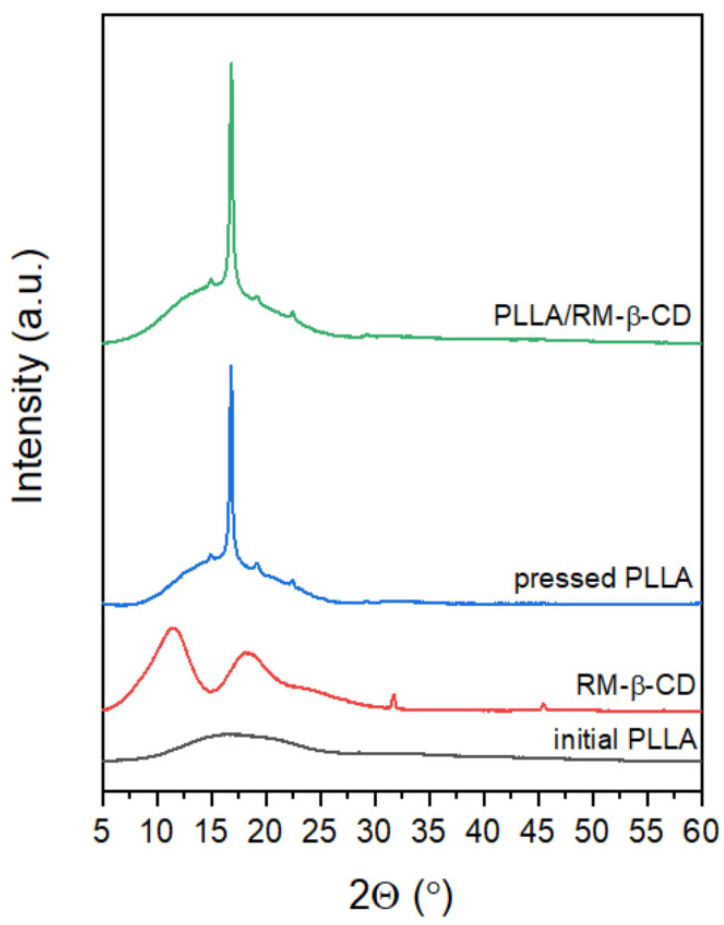
X-ray diffractograms of initial PLLA rigid film, RM-β-CD powder, and pressed PLLA and pressed PLLA/RM-β-CD films. All scans were normalized to maximal intensity of initial PLLA.

**Figure 7 ijms-23-07693-f007:**
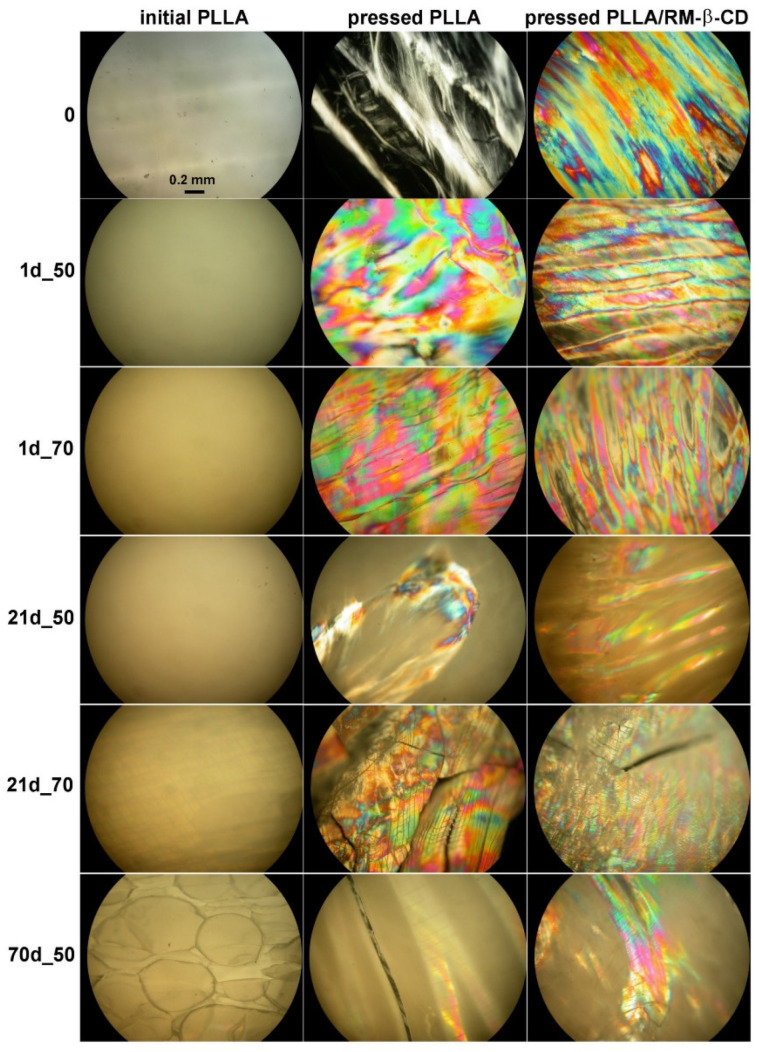
Representative photomicrographs of optical textures of the non-mesogenic thermoplastic initial PLLA rigid film, nematic mesophase of the pressed PLLA film, and colored planar texture of the chiral nematic mesophase of the pressed PLLA/RM-β-CD film before and after 1, 21, and 70 days of degradation at 50 and 70 °C (crossed polarizers, 25 °C, 100×).

**Figure 8 ijms-23-07693-f008:**
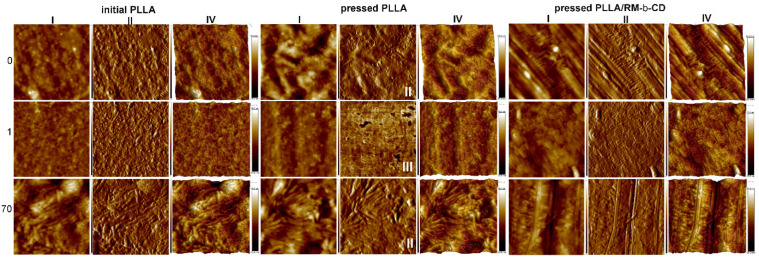
Representative AFM images with a scan size of a 1 × 1 μm height sensor (**I**), amplitude error (**II**), phase (**III**), and 3D (**IV**) images of the initial PLLA rigid film, pressed PLLA films obtained at a pressure of 5 tons for 1 min at 110 °C without (thread-like texture), and with 0.5 wt% of RM-β-CD (colored planar texture) before (0) and after 1 and 70 days of degradation at 50 °C.

**Figure 9 ijms-23-07693-f009:**
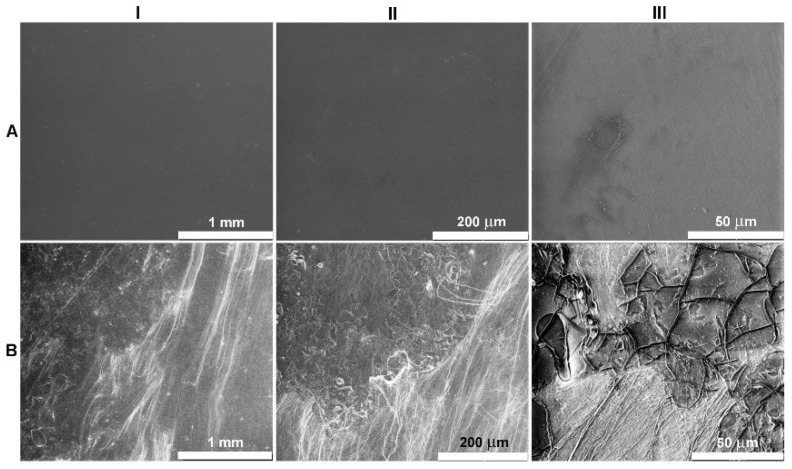
SEM images with a scan size of 3 mm × 2.6 mm (**I**), 600 μm × 520 μm (**II**), and 147 μm × 127 μm (**III**) of pressed PLLA (**A**) and PLLA/RM-β-CD (**B**) films.

**Table 1 ijms-23-07693-t001:** Molar mass and molar-mass dispersity of the initial PLLA rigid film and pressed PLLA and PLLA/RM-β-CD films, before and after 3, 7, 21, and 70 days of degradation at 50 and 70 °C.

Sample	Initial PLLA		Pressed PLLA		Pressed PLLA/RM-*β*-CD	
	50 °C	70 °C	50 °C	70 °C	50 °C	70 °C
**Before degradation**						
*M_w_* [g·mol^−1^]	179,000		177,000		184,000	
*M_w_/M_n_*	2.0		2.1		2.3	
**3rd day**						
*M_w_* [g·mol^−1^]	N/A	28,000	N/A	46,000	N/A	45,000
*M_w_/M_n_*	N/A	3.8	N/A	3.0	N/A	3.5
**7th day**						
*M_w_* [g·mol^−1^]	136,000	7,000	157,000	16,000	159,000	17,000
*M_w_/M_n_*	4.5	1.9	2.4	3.3	4.5	3.3
**21st day**						
*M_w_* [g·mol^−1^]	113,000	N/A	116,000	N/A	119,000	N/A
*M_w_/M_n_*	2.0	N/A	2.8	N/A	2.0	N/A
**70st day**						
*M_w_* [g·mol^−1^]	20,000	N/A	29,000	N/A	21,000	N/A
*M_w_/M_n_*	4.0	N/A	3.2	N/A	3.0	N/A

N/A—not available, *M_w_*—mass-average molar mass, *M_w_*/*M_n_*—molar-mass dispersity.

**Table 2 ijms-23-07693-t002:** Calorimetric parameters of the initial PLLA rigid film and pressed PLLA and PLLA/RM-β-CD films, before and after 3, 7, 21, and 70 days of degradation at 50 and 70 °C (I-heating run and for *T_g_* and Δ*c_p_* II- heating run with rate: 20 °C·min^−1^).

Sample	Initial PLLA		Pressed PLLA		Pressed PLLA/RM-*β*-CD	
	50 °C	70 °C	50 °C	70 °C	50 °C	70 °C
**Before degradation**						
*T_g_* [°C]	60.0		58.9		62.1	
Δ*c_p_* [J/g °C]	0.59		0.57		0.48	
*T_cc_* [°C]	125.6		126.2		122.0	
Δ*H_cc_* [J/g]	−8.08		−4.00		−9.17	
*T_m_* [°C]	152.4		148.9		150.3	
Δ*H_m_* [J/g]	8.8		7.91		13.77	
**21st day**						
*T_g_* [°C]	58.8	45.7	58.6	48.0	59.5	47.9
Δ*c_p_* [J/g°C]	0.60	0.59	0.24	0.55	0.60	0.58
*T_cc_* [°C]	108.8	-	97.6	-	110.5	95.0
Δ*H_cc_* [J/g]	−30.13	*-*	−12.65	-	−19.25	−8.28
*T_m_* [°C]	147.8/152.2	135.1	154.5	134.9/139.6	156.6	139.0
Δ*H_m_* [J/g]	31.11	58.04	12.70	61.88	19.45	55.73
**70th day**						
*T_g_* [°C]	51.8	0.5/23.2	54.7	−4.4/39.0	54.6	−11.8/38.9
Δ*c_p_* [J/g°C]	0.51	0.05/0.59	0.56	0.11/0.48	0.54	0.04/0.50
*T_cc_* [°C]	-	-	96.2	-	111.1	-
Δ*H_cc_* [J/g]	-	*-*	−5.28	-	−3.42	-
*T_m_* [°C]	132.8/149.8	91.4	152.3	81.2/99.5	153.0	82.8/112.3
Δ*H_m_* [J/g]	39.43	52.42	36.68	46.31	32.53	38.54

N/A—not available, *T_g_*—glass transition temperature from II-heating run after RC, Δ*c_p_*—the increment of heat capacity at the glass transition, *T_m_*—melting temperature, Δ*H_m_*—melting enthalpy, *T_cc_*—maximum of the exothermic peak of the cold crystallization temperature, Δ*H_cc_*—cold crystallization enthalpy.

**Table 3 ijms-23-07693-t003:** Lattice parameters, crystallite size, lattice strain, and residual stress determined for pressed PLLA^24^ and PLLA/RM-β-CD films, and pressed PLLA films, after 1 day of degradation at 50 °C (1d_50) and 70 °C (1d_70), and initial PLLA rigid film after 70 days of degradation at 50 °C (70d_50).

Sample	Lattice Parameters [Å]			Lattice Volume [Å^3^]	Crystallite Size [nm]	Lattice Strain [%]	Residual Stress [MPa]
	a	b	c				
reference polylactide ^1^	10.84	6.19	28.95	1942.5			
Pressed PLLA^24^	10.72	6.21	29.16	1940.8	41 ± 8	0.05 ± 0.01	4.8 ± 1.0
Pressed PLLA/RM-*β*-CD	10.67	6.19	28.41	1875.9	30 ± 1	0.91 ± 0.44	−2.8 ± 0.7
Pressed PLLA_1d_50	10.73	6.19	28.56	1897.5	18 ± 3	0.50 ± 0.06	6.7 ± 0.9
Pressed PLLA_1d_70	10.84	6.20	29.00	1948.8	25 ± 6	0.46 ± 0.06	13.8 ± 1.7
Initial PLLA_70d_50	10.64	6.13	28.94	1889.6	66 ± 6	0.05 ± 0.01	5.7 ± 0.6

^1^ data from International Centre for Diffraction Data (ICDD #00-064-1623).

**Table 4 ijms-23-07693-t004:** Roughness of initial PLLA rigid film and pressed PLLA and PLLA/RM-β-CD films, before and after 1 and 70 days of degradation at 50 °C.

Sample	Initial PLLA	Pressed PLLA	Pressed PLLA/RM-β-CD
**Before degradation**			
*R_q_* [nm]	1.3 ± 0.2	3.3 ± 0.3	3.9 ± 0.4
*R_a_* [nm]	1.0 ± 0.2	2.6 ± 0.3	3.1 ± 0.4
Image Z range * [nm]	7.7 ± 1.4	19.7 ± 3.0	28.5 ± 9.4
**1 day**			
*R_q_* [nm]	1.0 ± 0.1	1.5 ± 0.3	2.4 ± 0.7
*R_a_* [nm]	0.8 ± 0.1	1.2 ± 0.3	1.9 ± 0.5
Image Z range * [nm]	7.2 ± 0.5	10.8 ± 2.8	17.3 ± 6.4
**70 days**			
*R_q_* [nm]	2.4 ± 0.8	2.8 ± 0.1	3.4 ± 0.3
*R_a_* [nm]	1.9 ± 0.7	2.1 ± 0.1	2.6 ± 0.1
Image Z range * [nm]	18.1 ± 1.9	22.7 ± 4.7	28.1 ± 9.5

* Image Z range corresponds to the maximum roughness.

## Data Availability

The raw/processed data required to reproduce these finding is available upon request.

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
