# Peer review of "Nematic-to-Isotropic Phase Transition in Poly(L-Lactide) with Addition of Cyclodextrin during Abiotic Degradation Study"

_ijms, 2022, doi:10.3390/ijms23147693_

Round 1

Reviewer 1 Report

In the manuscript titled “Nematic-to-isotropic phase transition in poly(L-lactide) with addition of cyclodextrin during abiotic degradation study” by Joanna Rydz et al., the phase behavior of polymer materials based on poly(L-lactide) and cyclodextrin is considered. As the authors claim that non-mesogenic thermoplastic polymers can form “a chiral nematic mesophase during pressing, regardless of temperature and time”.

The comprehensive experimental study really shows an unusual phase behavior at which the phase with a strong optical birefringence typical for liquid crystals appears. Unfortunately, I cannot agree that the liquid-crystalline phase is formed.

My comments are as follows:

1.      Line 94 “In LCs, on the other hand, although the molecules still do not have fixed positions and remain unordered, like a liquid on its horizontal position, molecules can be ordered, in the form of a crystal along a vertical direction (anisotropy).”

As I right understand is means a translational ordering along a one direction. In contrast, here is a definition from [D. Demus, J. Goodby, G. W. Gray, H.-W. Spiess, and V. Vill, editors , Physical Properties of Liquid Crystals (Wiley-VCH Verlag GmbH, 1999).]:

“Liquid crystal state - -recommended symbol LC - a mesomorphic state having long range orientational order and either partial positional order or complete positional disorder”.

This indicates that LCs cannot be “in the form of a crystal” even along a one direction.

2.      Line 255 “The coexistence of domains with a colored planar texture (Figure 3B) and a schlieren texture of the N* mesophase (Figure 3C–F) was observed especially at the edges of the films, where the RM-β-CD distribution could be more uneven due to more difficult distribution”.

Line 370 “During degradation of pressed PLLA film with thread-like texture [24], especially after 1 day of degradation, a colored planar Grandjean texture of an N* mesophase was observed in POM, indicating that the degradation products trapped in the polymer matrix acted as a trigger for the N* mesophase”.

Actually, the obtained textures resemble an stressed polymer film rather than a nematic mesophase. If any polymeric film is elongated in a one direction, its structure becomes anisotropic due to the appearance of partial ordering of molecular chains (and optical anisotropy as a consequence). It is very typical for polymer materials which are used in everyday life [S. M. Crawford and H. Kolsky, Stress Birefringence in Polyethylene, Proc. Phys. Soc. Sect. B 64, 119 (1951). D. S. Ryu, T. Inoue, and K. Osaki, A Birefringence Study of Polymer Crystallization in the Process of Elongation of Films, Polymer (Guildf). 39, 2515 (1998)]. Such materials are not usually considered as liquid-crystalline, because their anisotropy is thermodynamically unstable.

3.      Line 314 “Performed analysis confirmed that the obtained materials exhibited both crystalline and nematic behavior”.

LC phase is the thermodynamically stable state of matter which differs from others. These results can indicate the coexistence of crystal phase and amorphous one. It is impossible for the material to exhibit both crystalline and nematic behavior. 

In conclusion, I strongly recommend to revise the interpretation of the obtained results. The study performed is actual and very interesting, but it requires a more accurate description.

Author Response

Answer to reviewers

We would like to express our gratitude to the reviewers and to acknowledge their effort to improve the paper's quality. Manuscript changes were highlighted in yellow (pdf file).

Reviewer 1

Point 1:

Line 94 “In LCs, on the other hand, although the molecules still do not have fixed positions and remain unordered, like a liquid on its horizontal position, molecules can be ordered, in the form of a crystal along a vertical direction (anisotropy).”

As I right understand is means a translational ordering along a one direction. In contrast, here is a definition from [D. Demus, J. Goodby, G. W. Gray, H.-W. Spiess, and V. Vill, editors , Physical Properties of Liquid Crystals (Wiley-VCH Verlag GmbH, 1999).]:

“Liquid crystal state - -recommended symbol LC - a mesomorphic state having long range orientational order and either partial positional order or complete positional disorder”.

This indicates that LCs cannot be “in the form of a crystal” even along a one direction.

Response 1:

During the linguistic revision, the sentence was changed and lost its original meaning, which is why this error has crept in. Thank you for your consideration. The sentence has been corrected using IUPAC phrasing.

Point 2:

Unfortunately, I cannot agree that the liquid-crystalline phase is formed.

Line 255 “The coexistence of domains with a colored planar texture (Figure 3B) and a schlieren texture of the N* mesophase (Figure 3C–F) was observed especially at the edges of the films, where the RM-β-CD distribution could be more uneven due to more difficult distribution”.

Line 370 “During degradation of pressed PLLA film with thread-like texture [24], especially after 1 day of degradation, a colored planar Grandjean texture of an N* mesophase was observed in POM, indicating that the degradation products trapped in the polymer matrix acted as a trigger for the N* mesophase”.

Actually, the obtained textures resemble an stressed polymer film rather than a nematic mesophase. If any polymeric film is elongated in a one direction, its structure becomes anisotropic due to the appearance of partial ordering of molecular chains (and optical anisotropy as a consequence). It is very typical for polymer materials which are used in everyday life [S. M. Crawford and H. Kolsky, Stress Birefringence in Polyethylene, Proc. Phys. Soc. Sect. B 64, 119 (1951). D. S. Ryu, T. Inoue, and K. Osaki, A Birefringence Study of Polymer Crystallization in the Process of Elongation of Films, Polymer (Guildf). 39, 2515 (1998)]. Such materials are not usually considered as liquid-crystalline, because their anisotropy is thermodynamically unstable.

Response 2:

The nematic phase has been confirmed by three techniques together (POM, DSC and X-ray) and the main evidence of mesogenic behavior is the observation of phase transitions as a function of temperature under POM. Heating helps to confirm the occurring effects as LC and not polycrystals or influence of stress, for example. The resulting enantiotropic mesophase was stable both at ambient temperature and during degradation at 50 °C as well as having a low energy state.

In addition, this LC phenomenon in polymers as consequence of pressing was known and described earlier in patent by Mathiowitz et al. Liquid crystalline polymers. WO2001068745A2, 2001 and in our earlier study (Janeczek et al. Poly(L-lactide) liquid crystals with tailor-made properties towards a specific nematic mesophase texture. ACS Sustainable Chem. Eng. 2022, 10, 10, 3323–3334).

Point 3:

Line 314 “Performed analysis confirmed that the obtained materials exhibited both crystalline and nematic behavior”.

LC phase is the thermodynamically stable state of matter which differs from others. These results can indicate the coexistence of crystal phase and amorphous one. It is impossible for the material to exhibit both crystalline and nematic behavior. 

In conclusion, I strongly recommend to revise the interpretation of the obtained results. The study performed is actual and very interesting, but it requires a more accurate description.

Response 3:

The unfortunate sentence has been removed. The word crystallinity has also been replaced with orderliness which is more accurate.

The point here is that the LC regions can cover whole or only a part of the material. Earlier investigations [Janeczek et al. Poly(L-lactide) liquid crystals with tailor-made properties towards a specific nematic mesophase texture. ACS Sustainable Chem. Eng. 2022, 10, 10, 3323–3334] had shown that the material was amorphous when processed up to 100 °C. The material preparation temperature between 110 °C to 130 °C (as for pressed PLLA and PLLA/RM-β-CD films) accelerated order making the material heterogeneous. At the processing temperature of 140 °C, the pressed PLLA films no longer showed LC properties as they became crystalline.

Similar observations have already been published as in the case blue phase: crystalline BPI* and BPII*, as well as amorphous BPIII* (see Coles, H. Pivnenko, M. Liquid crystal ‘blue phases’ with a wide temperature range, Nature 2005, 436, 997).

Reviewer 2 Report

The paper shows LCs mesophases formation and transitions of a Polyester matrix, and the effects of randomly methylated b-cyclodextrin addition upon hydrolytic degradation. The manuscript is very well written and the topic is interesting, although slightly far from a concrete technological application. The work is still well within the scope of the journal.

Few points need attention prior publication:

-          Fig. 1 : the samples shown in the bottom images (21d_70) have a different shape in respect to the others. Were all the other images(by row) the same specimens? Why using a different shape sample to show degradation after 21 days at 70 C? Please clarify.

-          Table 1 : Having molar mass and DSC data in the same table makes it quite big and slightly difficult to read, potentially leading to confusion. Please separate the data in 2 different tables.

-          Fig. 3, 5, 7 and 9 are too small.

-          Fig.4 and relative discussion : the endothermic and exothermic peaks (heating and cooling) are very close to just a standard PLA crystals melting point. It feels like a big claim that “DSC studies confirmed that pressed PLLA/RM-β-CD possessed LC properties.” Please rephrase, clarify and expand to justify your argument better (PP 7-8).

-          PP 13 : The pore depth increase in the order of 10-20 nm may be the result of many things, including process conditions, especially considering that the films were preparing by a sort of solid state processing (110 C). It may still be related to LCs structures, but it’s hard to say.

Author Response

Answer to reviewers

We would like to express our gratitude to the reviewers and to acknowledge their effort to improve the paper's quality. Manuscript changes were highlighted in yellow (pdf file).

Reviewer 2

Point 1:

Fig. 1: the samples shown in the bottom images (21d_70) have a different shape in respect to the others. Were all the other images (by row) the same specimens? Why using a different shape sample to show degradation after 21 days at 70 C? Please clarify.

Response 1:

Each sample is prepared independently. When removed after specific times, it is left for analysis. No other shape specimens were used after 21 days. After 21 days of degradation (21d_70) the samples disintegrated and fell apart. That's why they have a different shape. The macrographic image was contrasted to better distinguish the sample from the background and relevant sentence has been added to the text.

Point 2:

Table 1 : Having molar mass and DSC data in the same table makes it quite big and slightly difficult to read, potentially leading to confusion. Please separate the data in 2 different tables.

Response 2:

As suggested by the reviewer, the data was divided into 2 tables.

Point 3:

Fig. 3, 5, 7 and 9 are too small.

Response 3:

As suggested by the reviewer, the figures were enlarged.

Point 4:

Fig.4 and relative discussion : the endothermic and exothermic peaks (heating and cooling) are very close to just a standard PLA crystals melting point. It feels like a big claim that “DSC studies confirmed that pressed PLLA/RM-β-CD possessed LC properties.” Please rephrase, clarify and expand to justify your argument better (PP 7-8).

Response 4:

The nematic phase has been confirmed by the POM and DSC techniques together, and the main evidence of mesogenic behavior is the observation of phase transitions as a function of temperature under POM. The combination of the temperatures during heating under POM with the DSC helps to confirm the occurring effects as LC. The relevant sentence has been added to the text.

Point 5:

PP 13 : The pore depth increase in the order of 10-20 nm may be the result of many things, including process conditions, especially considering that the films were preparing by a sort of solid state processing (110 C). It may still be related to LCs structures, but it’s hard to say.

Response 5:

Measurements were made on several different samples which should eliminate any differences in sample preparation.

Round 2

Reviewer 1 Report

The Authors improved the text of the manuscript, but some points remain unclear:

“Response 2:

The nematic phase has been confirmed by three techniques together (POM, DSC and X-ray) and the main evidence of mesogenic behavior is the observation of phase transitions as a function of temperature under POM.”

It would be better to add a discussion about the difference between nematic phase of the non-mesogenic polymer under study and stressed polymer films (see the previous review).

DSC shows a weak phase transition, but it does not confirm the existence liquid crystalline phase.

 The ordering of polymer chains detected by X-ray can be the same as for stressed polymers. The POM pictures resemble stressed polymer (just an example, https://en.wikipedia.org/wiki/File:Food_Polarization-Dierking.jpg) rather than the N* and N phases which schlieren texture usually contains fingerprint patterns, focal conic defects, point defects (for nematics) etc. [I. Dierking, Textures of Liquid Crystals (Wiley, 2003)].

POM images raise also many minor questions. For example:

1)      Line 247 “colored planar texture of the 247 chiral nematic mesophase of the pressed PLLA/RM-β-CD film (C), nematic mesophase with a heterogeneous surface (G),”

-          how do the textures of “nematic” (G) and “chiral nematic” (C) differ from each other? What is the cholesteric pitch of the PLLA/RM-β-CD material?

2)      Line 292 “the nematic mesophase lost birefringence and was transformed into an isotropic phase and the texture turned colorless (Figure 5B)”

-          if any material has no birefringence (isotropic phase), it should be completely dark in crossed polarizers.

Author Response

Answer to reviewers

We would like to express our gratitude to the reviewers and to acknowledge their effort to improve the paper's quality. Manuscript changes were highlighted in yellow (pdf file).

Reviewer 1

Point 1:

Response 2:

The nematic phase has been confirmed by three techniques together (POM, DSC and X-ray) and the main evidence of mesogenic behavior is the observation of phase transitions as a function of temperature under POM.”

It would be better to add a discussion about the difference between nematic phase of the non-mesogenic polymer under study and stressed polymer films (see the previous review).

DSC shows a weak phase transition, but it does not confirm the existence liquid crystalline phase.

 The ordering of polymer chains detected by X-ray can be the same as for stressed polymers. The POM pictures resemble stressed polymer (just an example, https://en.wikipedia.org/wiki/File:Food_Polarization-Dierking.jpg) rather than the N* and N phases which schlieren texture usually contains fingerprint patterns, focal conic defects, point defects (for nematics) etc. [I. Dierking, Textures of Liquid Crystals (Wiley, 2003)].

 Response 1:

Observations under POM eliminate all doubts as to the DSC spectra. A material in which a phase transition is observed as a function of temperature and forms a mesophase exhibits LC properties. The most important evidence of mesogenic behavior is the observation of phase transitions as a function of temperature under POM. A reversible nematic to isotropic transition is not observed in the case of a stressed polymer film.

A detailed discussion of this topic is provided in a previous publication, which described not only the LC phenomenon in polylactide but, also the formation of all textures. To duplicate this and the described behavior of the LC phase as a function of temperature would be to plagiarize of own publication in which this discussion was conducted, and thus the present publication often refers to the previous one.

As suggested by the reviewer, an appropriate sentence has additionally been added to the text.

Point 2:

POM images raise also many minor questions. For example:

Line 247 “colored planar texture of the 247 chiral nematic mesophase of the pressed PLLA/RM-β-CD film (C), nematic mesophase with a heterogeneous surface (G),”

how do the textures of “nematic” (G) and “chiral nematic” (C) differ from each other? What is the cholesteric pitch of the PLLA/RM-β-CD material?

Response 2:

The nematic mesophase, when viewed under a POM between crossed polarizers, created distinctive dark thread-like structures (Figures 1B and S1 in the Supporting Information in [25]) that are topological defects. Defects in LC systems are important for the identification of mesophase types.35 In the chiral nematic mesophase, the molecules twist perpendicular to the director axis (axis of rotation), with the molecular axis parallel to the director. The twist angle between adjacent molecules results from the asymmetric packing leading to a longer-range chiral order. The distance along the helical pitch for a full rotation of the mesogens is a strong function of the temperature. Generally, the helical pitch of cholesteric LCs is of the order of several hundred nanometers (the wavelength of visible light) and thus exhibits interference colors.6−8 In POM, the isotropic phase is dark under crossed polarizers, while the birefringent nematic mesophase exhibits interference colors. The bright colors are due to the difference in rotatory power resulting from domains with different cholesteric pitches.5 The schlieren texture of an N phase is observed for a flat sample between crossed polarizers, showing a network of black brushes connecting centers of point and line defects (Figures 1E,F and S4 in the Supporting Information in [25]).36,37 This texture is observed in a planar cell, where the director aligns parallel to the surface and is organized around point disclinations, surface disclination lines, and inversion walls.38

Additional references to the publication [25] have been added to the text.

Point 3:

Line 292 “the nematic mesophase lost birefringence and was transformed into an isotropic phase and the texture turned colorless (Figure 5B)”

if any material has no birefringence (isotropic phase), it should be completely dark in crossed polarizers.

Response 3:

The unfortunate word was used in the text meaning that the material had lost its color. The isotropic phase is dark under crossed polarizers [25] as it can be seen in the photomicrograph.

As suggested by the reviewer, an appropriate word has been added to the text.